# Optimization Design of the Landscape Elements in the Lhasa Residential Area Driven by an Orthogonal Experiment and a Numerical Simulation

**DOI:** 10.3390/ijerph19106303

**Published:** 2022-05-22

**Authors:** Lixing Chen, Yingzi Zhang, Zhengzheng Luo, Fei Yao

**Affiliations:** School of Architecture, Southwest Jiaotong University, Chengdu 610031, China; chenxingxing0511@163.com (L.C.); 2019200793@my.swjtu.edu.cn (Z.L.); 2019211241@my.swjtu.edu.cn (F.Y.)

**Keywords:** high altitude, landscape elements, outdoor thermal comfort, orthogonal experiment

## Abstract

Landscape elements have become an important means to improve the quality of life of residents because of their direct influence on the thermal environment, but the selection and configuration of landscape elements have different effects on human thermal comfort in different climate conditions. In this research, the typical residential area of Lhasa in Tibet was taken as the research object, the experimental scheme was prepared using an orthogonal test, and the simulation was carried out using ENVI-met to explore the influences of the green configuration, water area, and ground reflectance, as well as their interaction with the thermal environment in winter and summer under alpine climate conditions. Taking the physiological equivalent temperature (PET) as the optimization index, the optimal design scheme for the synergistic effect of the residential landscape elements was determined. The results were as follows. (1) The order of the landscape configuration factors was as follows: green configuration > water area > leaf area index > ground reflectance in summer. In winter, the order was green configuration > water area > ground reflectance > leaf area index (LAI). (2) With the combined driving of the orthogonal test and the numerical simulation, the optimal scheme of the landscape elements was determined, which was “tree shrub lawn, water area ratio 16%, ground reflectance 0.5, and LAI = 3 m^2^/m^3^”. (3) Finally, the optimal design strategy of the landscape configuration was proposed for the typical outdoor active space of the Lhasa residential area.

## 1. Introduction

### 1.1. Background

According to the sixth report of the Intergovernmental Panel on Climate Change (IPCC), global warming caused by human activities is 1.0 °C higher than before industrialization, global warming is continuing, and the risk of extreme weather events is further increasing [1,2]. The urban heat island effect (UHI) caused by climate change and anthropogenic changes in rapid urban development has directly led to the emergence of urban problems such as high temperature, drying, poor ventilation, and pollutant accumulation. These changes reduce the urban outdoor thermal environment and the thermal comfort of residents [3,4] and increase the energy consumption of urban buildings [5,6]. They also endanger human health, induce various health problems [7,8], and can even lead to death [9]. The China Climate Change Blue Book released by the China Meteorological Administration in 2021 pointed out that: “China is a sensitive area and a significant impact area of global climate change, and the heating rate is significantly higher than the global average in the same period [10]”. It has become crucial to take strong measures to improve the urban environment and improve the quality of the human living environment. There is already a large amount of research trying to find optimal solutions to mitigate the many challenges posed by the growing climate problem. From the perspective of urban planning, this is mainly reflected in two aspects: urban building layout and urban landscape (vegetation, water body, underlying surface, etc.). As a nature-based solution, urban landscapes are believed to play a key role in improving microclimates and mitigating heat island effects [11]. In addition to this, a good landscape environment can enhance residents’ well-being while improving residents’ comfort [12,13].

### 1.2. Literature Review

Determining how to create a healthy outdoor thermal environment based on nature-based solutions has become a hot research topic, and it is particularly important to improve the outdoor thermal environment in residential areas because they are the basis for many outdoor activities. The landscape environmental factors that are closely related to the outdoor thermal environment and thermal comfort of a residential area mainly include greening, water bodies, and underlying surface. In terms of greening, in hot and arid areas, choosing trees that can reduce solar radiation is the most effective urban design strategy for reducing physiological equivalent temperature (PET) [14]. The research of Middel et al. [15] proved that compact building layouts had a greater impact on air temperature during the day than greening, the use of trees in compact low-density building areas reduced the temperature by 2 °C, and trees in high-density building areas could effectively reduce air temperature by 0.2–0.4 °C [16], but this was ineffective in Cairo’s low-density built-up areas [17]. In hot and humid climates, studies by Srivanit et al. [18] and Zhuang et al. [19] confirmed that trees could affect solar radiation and change the wind environment, significantly reducing the average radiation temperature (Tmrt) and PET. The structure of the canopy, as well as the size, shape, and color of the leaves, affected the level of solar radiation [20]. In cold climates, in addition to improving the outdoor thermal environment in summer [21], trees can also effectively improve the outdoor wind environment in winter by adjusting the layout of trees [22]. In temperate climates, Lee et al. [23] confirmed that the combined form of trees and lawns had a more significant effect on Tmrt and PET than lawns alone. In terms of water bodies, Lu et al. [24] used a humid and hot climate as the background to confirm that a water body had a cooling effect in the daytime warming stage. Additionally, the evening cooling stage played a role in heat preservation and had the effect of humidifying and regulating the wind speed. In cold climatic conditions, the scale of influence of hot and humid environments in waterfront residential areas depends largely on the volume ratio, the height of the embankment, and the greening [25]. In addition, the characteristics of the ground are an important factor that affect the thermal comfort of residents [26]. In a humid and hot climate, the research of Kwan et al. [27], Guo et al. [28], and Song et al. [25] found that the temperature of an impermeable surface was higher than the temperature of a permeable surface, and a natural lower pad surface had a better cooling effect. In mild climates, Santamouris et al. [29] showed that materials with high reflectivity could significantly affect the thermal comfort of the human body, but the study of Taleghani et al. [30] showed that both the Tmrt and PET increased with increasing reflectivity.

Based on the above literature review, we found that the impact of landscape elements on the regional thermal environment in different climatic zones has been studied many times before, and that their research results were also different from each other, but studies of the synergy mechanism between various landscape elements are still relatively lacking. Each city is in a specific regional atmospheric environment, and the differences in urban climate affect the mechanism of improvement of landscape elements for the outdoor thermal environment. China’s Tibet region is affected by its special geographical location, climatic conditions, and religious culture. The impact of landscape elements on the thermal environment is different from those of other climatic zones. Local residents carry out a large number of outdoor activities every day, and frequent and long-term outdoor activities put forward higher requirements for the outdoor environment. However, there are relatively few studies on high-altitude cold climate zones, and it is difficult to provide theoretical support for improving the quality of the living environment in extremely cold climates. Therefore, in this study, in order to better understand the impact of residential landscape elements on the outdoor thermal environment and thermal comfort in a high-altitude cold climate area, taking Lhasa as an example, the thermal environment of a residential area with different landscape element configurations was numerically simulated based on the orthogonal experimental design method. The PET was taken as the optimization objective, and the simulation results were analyzed with intuitive analysis and variance analysis. The main objectives were as follows:(1)To clarify the primary and secondary effects and advantages of various landscape elements on an outdoor thermal comfort under high altitude and cold climate conditions.(2)To clarify the synergistic relationship between landscape elements and their interaction with outdoor thermal comfort under high altitude and cold climate conditions.(3)To put forward the experimental scheme and strategy of landscape element allocation optimization in a residential area for the conditions of high altitude and a cold climate.

## 2. Research Methodology

### 2.1. Climate Conditions and Residence in Lhasa

Lhasa is located in the middle of the Qinghai Tibet Plateau, with an altitude of 3650 m. The plane of the city of Lhasa is distributed in a belt from east to west. Two mountains (South Mountain and North Mountain of Lhasa) sandwich the city, and a river (Lhasa River) passes through the city. It is a typical axial development city [31], as shown in Figure 1. According to the Chinese standard meteorological data, the hottest month is June, and the coldest month is January. The monthly average maximum temperature is above 20 °C, the monthly average minimum temperature is below −3 °C, and the annual temperature difference is 18.1 °C [32]. Lhasa belongs to the cold region in China’s architectural climate zoning, but because it is located in the Qinghai Tibet Plateau, the climate conditions of Lhasa are significantly different from those of Beijing, Xi’an, Lanzhou (cold climate region), and other cities. By comparing the enthalpy and humidity diagrams of the three cities (Figure 2), it can be seen that the annual temperature and the humidity of Lhasa are relatively lower, and the annual temperature difference is also relatively smaller. Beijing and Xi’an also have coexisting heat and humidity, but Lhasa has a dry climate, so there is no such phenomenon.

In this research, 60 residential areas were investigated and analyzed based on the following two factors (Table 1 and Table 2). (1) Architectural layout: most of the architectural layout forms were the determinant at high south and low north as well as the mixed type of peripheral and determinant combination, and the number of building floors was mainly that for multilayer and small high-rise buildings. (2) The current situation of the landscape element allocations in residential areas, which is as follows: The common greening forms of green space are trees, shrubs, and grass. The types of underlying surfaces mainly include floor tiles, permeable bricks, gravel roads, grass planting bricks, permeable plastic, and lawns. The number of waterscape layout areas is small, the waterscape form is relatively single, and the shapes are mostly still water and fountains.

### 2.2. Characteristics of Residents’ Behavior Habits

The Lhasa area is affected by strong radiation, and the demand for shade space in summer is greater than that in winter. There is also a greater objective demand for the comfort of the outdoor space thermal environment in summer and winter (Figure 3). In this study, through a questionnaire survey, we examined the behavioral characteristics of outdoor activities of the local residents in Lhasa. The content of the questionnaire mainly included two parts: (1) basic information about respondents (gender, age, name, and ethnicity); and (2) respondents’ behavior habits (activity time, activity form, and preferred activity venue). Overall, 325 questionnaires were distributed in winter and summer, and 311 questionnaires were effectively recovered. The effective rate of the questionnaires was more than 94%. The statistical analysis questionnaire showed that the main types of activities of Lhasa residents in the residential area were dancing, walking, and fitness. The main activities of residents in summer were concentrated at the times of 8:00–10:00 and 18:00–20:00. Residents were more likely to move under the lawn, waterscape, and shade of trees. The main activities in winter were at the times of 9:00–11:00 and 15:00–17:00. Residents were more likely to participate in activities next to shadeless squares and fitness equipment (Figure 4).

### 2.3. Research Framework

The research framework of this study is shown in Figure 5: (1) The landscape elements model of the residential area was grouped with the orthogonal design method to determine the experimental scheme. (2) ENVI-met, a three-dimensional urban microclimate simulation software jointly developed by Michael Bruse and Heriberfleet of Bochum University in Germany, was used for numerical simulation. This study used ENVI-met version v4.4.3 (Michael Bruse, Mainz, Germany) for the calculation to study the influence degree and the order of different landscape elements and their interaction with the thermal environment in residential areas. (3) The experiment optimization scheme with the collaborative action of landscape elements in a residential area was determined by taking the PET as the test index, and finally, the optimal design strategy was put forward.

### 2.4. Orthogonal Experimental Design

#### 2.4.1. Establishing Standard Model of a Residential Area

According to previous survey data, in this research, the residential area model of parallel north–south layout was set, with a single building height of six stories, a total building height of 18 m, a building surface width of 70 m, and a depth of 15 m. The space between the buildings was 24 m in the north–south direction and 6 m in the east–west direction, and the width of the roads around the residential area was 10 m. Considering the influence of the boundary of the building model on the simulation area, half blocks were added around the simulation center. The entire simulation area was 250 m × 150 m × 38 m, the number of grids was 50 × 50 × 38, and the unit grid size was 5 m × 3 m × 1 m. The standard model was initially established, as shown in Figure 6a. The two buildings in the middle of the basic model were replaced by a 54 m × 70 m open space, and the landscape elements in the area were changed to study the impact of different landscape elements on the microclimate (Figure 6b).

#### 2.4.2. Determination of Simulation Factors and Level of Orthogonal Experiment

With the premise that the green area of the public green space was not less than 70% [34], A stands for a different greening configuration; B stands for different water areas, with a water depth of 1.75 m by software default, arranged in the middle; and C stands for different underlying surface reflectances. In this research, we studied the impact of the tree leaf area index (LAI) on the outdoor thermal environment. According to the literature [35], the LAI of most plants is between 0–6 m^2^/m^3^. Therefore, we set the LAI to 1, 3, and 5. Thus, D then stands for different LAIs based on the JX-A1 condition. The factors and their level simulation conditions are shown in Table 3. Based on the four factors and their three levels determined in Table 3, the factor level and the plane diagram of the orthogonal test were established (Table 4).

#### 2.4.3. Preparation of Test Scheme

In this research, an orthogonal standard table L27 (3^13^) was selected to prepare the test plan. There were 81 comprehensive test schemes for the four-factor, three-level model. After the orthogonal test method was used, only 27 representative working conditions for Table 5 needed to be simulated.

#### 2.4.4. Determination of Simulation Time and Boundary Conditions

In order to ensure the universality and practical significance of the research, a typical simulation day that could best reflect the special climate of Lhasa was selected and the meteorological conditions of the initial simulation were determined. The typical meteorological days were June 21 in summer and January 21 in winter. Since the wind speed data in the special meteorological data set for the thermal environment analysis of Chinese buildings were collected only four times a day, the wind speed values in June and January were averaged as the simulated initial wind speed. The wind direction was the high-frequency wind direction of the current month. The setting of the specific boundary conditions is shown in Table 6.

## 3. Numerical Simulation Results and Optimization

PET is a thermal environment evaluation index based on Memi (the Munich energy balance model) [37]. Rayman software was used to calculate the PET. This software could correct for a low-pressure and high-altitude climate environment, such as the Lhasa residential area. According to the behavior characteristics of local residents, the hours of 8:00–10:00, 15:00–17:00, and 18:00–20:00 in summer and 9:00–11:00 and 15:00–17:00 in winter were selected for the simulation result analysis. The ENVI-met simulation software program was used for the simulation. The parameters of the air temperature, relative humidity, wind speed, and average radiation temperature were imported into the Rayman software to calculate the PET value. The physiological equivalent temperature level table (Table 7) was used to judge the thermal comfort status of each working condition. The comfort status of the human body is in the range of 18 °C to 23 °C. Furthermore, the primary and secondary effects and the optimal level of each factor were analyzed with the visual analysis method. Then, the significance of each factor and its interaction with the test index (PET) was determined using analysis of variance, and the contribution of each factor to the thermal comfort was comprehensively evaluated.

### 3.1. Primary and Secondary Influence of Factors and Ranking of Excellent Level

The PET values of the four factors at three levels and their average values were calculated in order to draw the PET trend diagram (Figure 7). The results of the orthogonal test are shown in Table 8. Based on the overall trend of the PET, in summer, the PET values of each factor were in a slightly cool range in the morning, which had little impact in the morning, and in a warm and slightly warm range in the afternoon and evening, which had a greater impact in the afternoon and evening. Comparing the range R of each factor in the morning, afternoon, and evening, we could see that the order of influence was greening configuration > water area > leaf area index > ground reflectance. In winter, the PET values of all factors were in the very cold range in the morning and in the slightly cool range in the afternoon. By comparing the range R of each factor in the morning and afternoon, we could see that the order of influence was greening configuration > water area > ground reflectance > leaf area index.

The level indexes of each factor, K1, K2, and K3, and the average values K1, K2, and K3, were compared, and the ranking of the superior level of each factor was analyzed. In summer, the comfortable range of the PET was selected as the optimal target: (1) For the greening configuration, the combination mode of tree + lawn and shrub + lawn was the best in the morning. The combination mode of tree + shrub + lawn was the best in the afternoon and evening, and the order of its level was tree + shrub + lawn > tree + lawn > shrub + lawn. (2) The ground reflectance of 0.5 was the best in the morning, and that of 0.3 was the best in the afternoon and evening. The order of the levels was ground reflectance 0.3 > ground reflectance 0.5 > ground reflectance 0.2. (3) The proportion of water area was negatively correlated with the PET, and the PET decreased with increasing water area. (4) For the leaf area index of the trees, LAI = 1 m^2^/m^3^ in the morning was the best, LAI = 5 m^2^/m^3^ in the afternoon and evening was the best, and the order of the optimal levels was LAI = 5 m^2^/m^3^> LAI = 3 m^2^/m^3^ > LAI = 1 m^2^/m^3^. The results showed that: (1) The tree, shrub, and lawn, and shrub + lawn combination modes were the best in the morning, the tree + shrub + lawn combination mode was the best in the afternoon, and the tree + lawn combination mode was the worst in winter. Considering the greater demand for sunshine space in the Lhasa area in winter, the comprehensive ranking of its level was tree + shrub + lawn > shrub + lawn > tree + lawn. (2) The ground reflectance of 0.5 in the morning and afternoon was the best, and the order of its levels was ground reflectance 0.5 > ground reflectance 0.3 > ground reflectance 0.2. (3) The water area proportion was negatively correlated with the PET, and the PET decreased with the increase of the water area. (4) The LAI = 3 m^2^/m^3^ in the morning and afternoon was the best, and the order of the best levels was LAI = 3 m^2^/m^3^ > LAI = 1 m^2^/m^3^ > LAI = 5 m^2^/m^3^.

### 3.2. Analysis of Influence Degree and Significance of Factors

The statistical product service solutions (SPSS) statistical analysis software from IBM was used to analyze the significance of each influencing factor. The version of IBM SPSS statistics 22 was adopted in this research. The analysis of variance is shown in Table 9. The larger the F value was, the more significant the influence on the test index was. The Sig. Value was the significant index of the test. When the significant level was 0.05, Sig. < 0.05, indicating that the influence factor had a significant influence on the test index [38].

On a summer morning, the Sig. values were less than 0.05, except for the interaction between the green configuration and the ground reflection and the interaction between the water area and the ground reflection, which had a significant impact on the PET of the outdoor public activity space in the residential area. According to the F value, the significant order of the influencing factors was green configuration > water area > ground reflectivity > leaf area index of trees > interaction between green configuration and water area. The influence of the factors on the test indexes was significantly enhanced in the afternoon and evening, and the interaction between the greening configuration and the ground reflectance was not significant. The significant order of the influencing factors was greening configuration > water area > leaf area index of trees > ground reflectance > interaction between surface reflectance and water area > interaction between greening configuration and water area.

In winter, except for the interaction between the greening configuration and the ground reflectance, the Sig. values were less than 0.05 in the morning, which had a significant impact on the PET in the outdoor public activity space. According to the F value, the significant order of the influencing factors was greening configuration > water area > ground reflectivity > leaf area index of trees > interaction between water area and ground reflectance > interaction between greening configuration and water area. In the afternoon, the order of influencing factors was greening configuration > water area > ground reflectivity > leaf area index of trees > interaction between ground reflectance and water area.

The contribution rate of each factor ρ_j_, and its interactions, are shown in Table 8. The results were as follows: (1) The effect of the green configuration on the PET was higher in winter and summer. (2) The influence of the water area on PET increased with time. (3) The influence of the ground reflectance and the leaf area index of the trees on the PET was lower than that of the other two factors in summer; the influence of ground reflectance on the PET was greater in the morning in winter. Compared with the other three factors, the leaf area index of the trees had less influence on the PET. (4) In the comparison of the three kinds of interaction, for the interaction of the water area and ground reflectance, the green configuration had an obvious effect on the improvement of the PET, which was greater in the morning in summer and winter, and the interaction between the green configuration and the ground reflectance was relatively small.

In general, the green configuration and the water area in winter and summer played a decisive role in the thermal environment of the outdoor activity space in the residential area. In summer, the improvement effect of the water area gradually increased with time, and the improvement effect of the ground reflectance and leaf area index of the trees was weaker than the other two factors. In winter, the influence of the ground reflectance and LAI of the trees on the PET was weaker than that in summer. Among the three interactions, the interaction of the water area with the green configuration and the ground reflectance had a more obvious influence on the PET in summer, the interaction of the water area with the ground reflectance had the most obvious influence on the PET in winter, and the interaction between the green configuration and the ground reflectance had less influence in winter and summer. Therefore, in the selection of the best combination of interaction factors in future research, the interaction between water area, greening configuration, and ground reflectance should be considered.

### 3.3. Analysis of The Best Combination of Factors Interaction

It can be seen from the above that the factors with significant interactions were the water area and the greening configuration, as well as the water area and the ground reflectance, i.e., A and B, as well as B and C. The calculation results for the interaction of the PET average values corresponding to each factor level in winter and summer and the three periods are shown in Table 10. The results showed that the combination of greening configuration and water area in summer was the best, A2B1 in the morning was the best, and AiB1 had higher thermal comfort than others. In the afternoon, A3B1 was the best, and the combination of AiB1 and A3Bi had higher thermal comfort than others. In winter, the best combination of water area and ground reflectance was B1C3 in the morning. BiC3 and B1Ci had higher thermal comfort than others. B1C3 in the afternoon was the best, and B1Ci had higher thermal comfort than others (I = 1, 2, 3).

### 3.4. Optimization Results of Test Scheme

The weighted averages of 27 test schemes were calculated and sorted according to the advantages and disadvantages of the test schemes. The optimization results are shown in Table 11. The terms PET_s_ and PET_w_ represent the weighted average calculation values of the PET in summer and winter, respectively. By comparing the ratio of the PET in winter and summer, the optimal allocation scheme of the outdoor activity space in the residential area in winter and summer was comprehensively evaluated. The smaller the ratio was, the better the experimental scheme was in winter and summer. The plane diagram for winter and summer and the comprehensive optimal scheme are shown in Figure 8. The results showed that the optimal test scheme was 26 in summer, that is, “tree shrub lawn, water area 50%, ground reflectance 0.3, LAI = 5 m^2^/m^3^”; the optimal test scheme was 21 in winter, that is, “tree-shrub-lawn, water area 16%, ground reflectance 0.5, LAI = 3 m^2^/m^3^”; the optimal test scheme for two seasons was 21. 

Therefore, the optimization results for the summer and winter test schemes were as follows: (1) For the combination of tree + shrub + lawn, the greening configuration was the best in the two seasons. The planting of trees and shrubs should follow the principle of “shading in summer and wind prevention in winter”. Planting tall deciduous trees on the west side of the activity site could not only cause the activity space to have more sunshine in the morning, but also form the shade space in the afternoon or evening. (2) For the water area, the appropriate area of the water body should be set up according to the needs of the users of the active space and with the premise of ensuring the scope of the activity. In addition, the interaction between the water body and the trees is significant, and the combination of the water body and the trees could enhance the improvement of the thermal environment. (3) For the ground pavement, it is advisable to choose lower-surface materials with relatively high reflectivity and high permeability, such as permeable brick, grass planting brick, or a light-colored granite floor. (4) The LAI of trees was not suitable to be too large or too small, which could meet the requirements of shade in summer and sunshine in winter.

## 4. Discussion

Our research showed that different landscape elements not only had great differences in their action mechanisms in thermal environments but also had some differences in the synergy mechanism between various elements. The results showed that in winter and summer, the greening configuration and water area played a decisive role in the thermal environment of the outdoor activity space in residential areas, and the improvement potential of the ground reflectance and LAI of tree was weaker than that of the other two factors. These results were confirmed in the studies of Li [39], Lu [24], and Zhuang [19]. In view of the superior level ranking of the impact of various influencing factors on the thermal comfort of outdoor space in residential areas in winter and summer for the condition of high-altitude cold climate, we concluded that “trees + shrubs + lawn”, “water area 33%”, “ground reflectance 0.3”, and “tree leaf area index LAI = 3 m^2^/m^3^” were the optimal levels of various landscape elements. This result was also confirmed by existing studies, but there were also some differences. For example, by studying the thermal acceptability of residents for different landscape element configurations, Li. confirmed that the setting of shrubs improved the outdoor space acceptance rate in winter but reduced the outdoor thermal acceptance rate when the solar radiation was strong in summer. High reflection ground reduced the heat acceptance rate of outdoor space in summer and improved the heat acceptance rate of outdoor space in winter [39]. The research of Santamouris et al. [29] showed that in a mild climate, materials with high reflectivity could significantly affect the thermal comfort of the human body. Other studies, such as the study of Zhang et al. [40], showed that trees had a large leaf area index and strong temperature regulation ability, which was also different from the conclusion of this study. It was speculated that the difference was due to the particularity of climate conditions in Lhasa; in particular, the strong solar radiation, long sunshine time, and large temperature difference between morning and evening. Therefore, high reflectivity ground and a large tree leaf area index would reduce the outdoor thermal comfort for some time periods. In addition, this study also proposed the impact of the interaction of various landscape elements on thermal comfort. There was an obvious interaction between the water area, greening configuration, and ground reflectivity. In the afternoon, when the thermal environment was relatively uncomfortable, the combination of arbor + shrub + lawn and different water areas always had higher thermal comfort in winter and summer. The water area and ground reflectance were different in winter and summer. The combination of 50% water area and different ground reflectance had higher comfort in summer, and the combination of 16% water area and different ground reflectance had higher comfort in winter. In winter, due to the low temperature in Lhasa, the water body was frozen, which also had an impact on the improvement of the thermal environment. The conclusions of Song et al. [25] were also verified. Finally, the optimal landscape element allocation scheme in winter and summer was: “arbor + shrub + lawn”, “water area of 16%”, “ground reflectance of 0.5”, and “tree leaf area index of 3 m^2^/m^3^”. However, the quantitative research on the configuration parameters of different types of landscape elements in previous research was relatively sparse and mainly focused on the single factor research of trees, water, hard ground, and so on. For example, some of the focus was on the form of greening layout [22], the morphological index of trees [20], the difference of tree species [41,42], and the shapes and layouts of water bodies [43]. Therefore, in order to better guide actual projects, the research on these topics is a key issue that needs to be considered and deepened in future research.

## 5. Conclusions

Through the analysis and discussion of the impact of the landscape element configuration of outdoor activity space on the thermal comfort in residential areas in Lhasa, the following three conclusions were drawn:(1)Green configuration was the key factor affecting the residential outdoor thermal environment, followed by the water area. The improvement potential of the tree leaf area index and ground reflectance on outdoor thermal comfort was relatively weak.(2)The interaction between the water area, greening configuration, and ground reflectivity affected the thermal comfort of outdoor public spaces in residential areas.(3)In winter and summer, the optimal landscape element allocation scheme for the outdoor activity space in the residential area of Lhasa for the conditions of high altitude and cold climate was as follows: “arbor + shrub + lawn”, “water area of 16%”, “ground reflectance of 0.5”, and “tree leaf area index of 3 m^2^/m^3^”.

Based on the above conclusions, the appropriate landscape element combination scheme was determined according to the functional requirements with the objectives of climate suitability, comfort, energy conservation, and functionality. The optimization scheme could directly refer to the optimized layout mode shown in Figure 8. There were still some limitations in this study because this study focused on the configuration of different landscape elements and lacked the consideration of trees, water form, and layout. These factors are also the problems we will focus on in the future.

## Figures and Tables

**Figure 1 ijerph-19-06303-f001:**
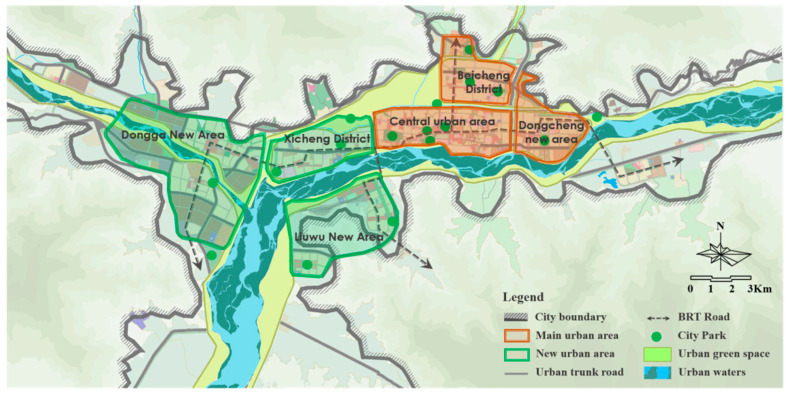
City map of Lhasa.

**Figure 2 ijerph-19-06303-f002:**
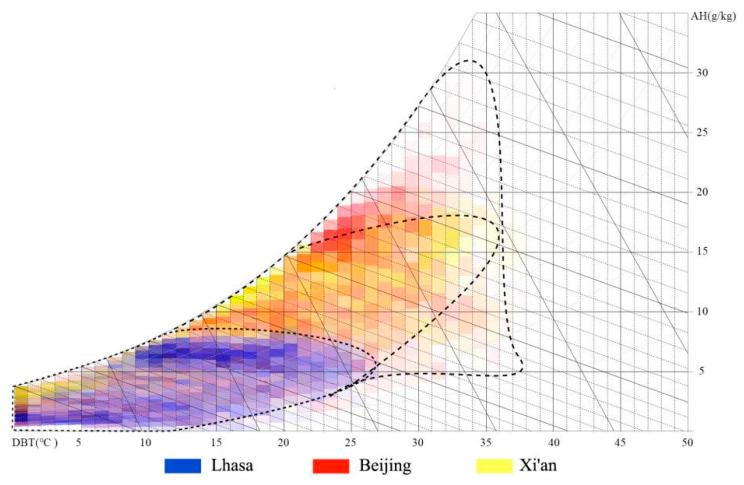
Enthalpy humidity diagram and solar radiation intensity comparison diagram of Lhasa, Beijing, and Xi’an (Adapted with permission from Ref. [33]. 2021, Chen, L.)

**Figure 3 ijerph-19-06303-f003:**
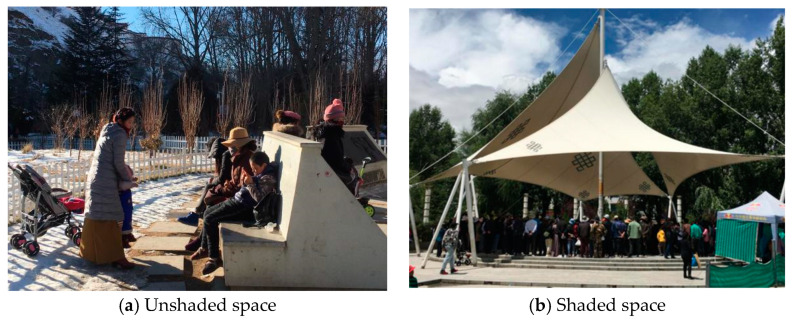
Outdoor environmental conditions of the Lhasa residential area.

**Figure 4 ijerph-19-06303-f004:**
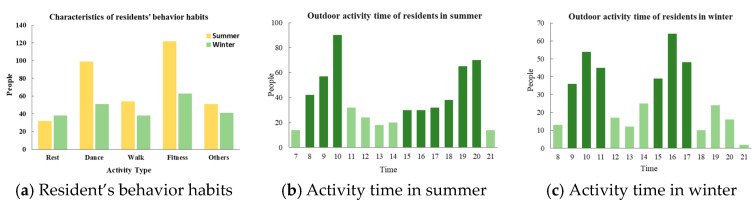
Characteristics of residents’ behavior habits.

**Figure 5 ijerph-19-06303-f005:**
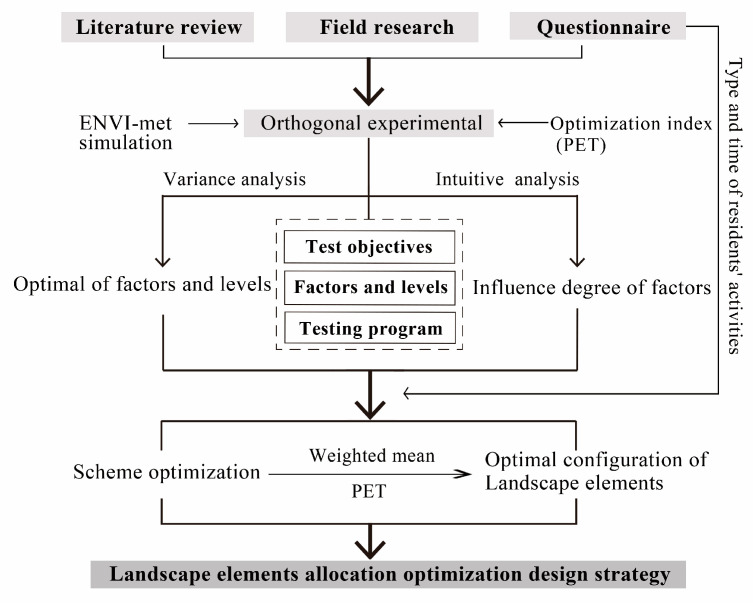
Research framework.

**Figure 6 ijerph-19-06303-f006:**
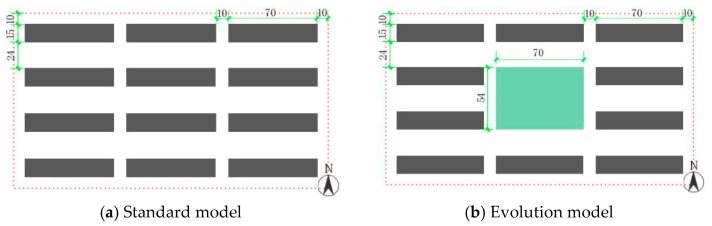
Standard model plan of residential area.

**Figure 7 ijerph-19-06303-f007:**
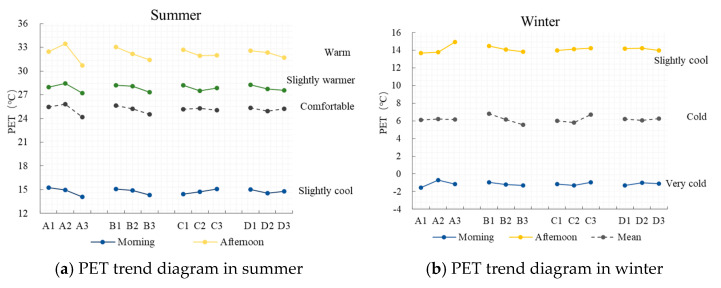
PET trend of outdoor public activities in residential areas in summer and winter.

**Figure 8 ijerph-19-06303-f008:**
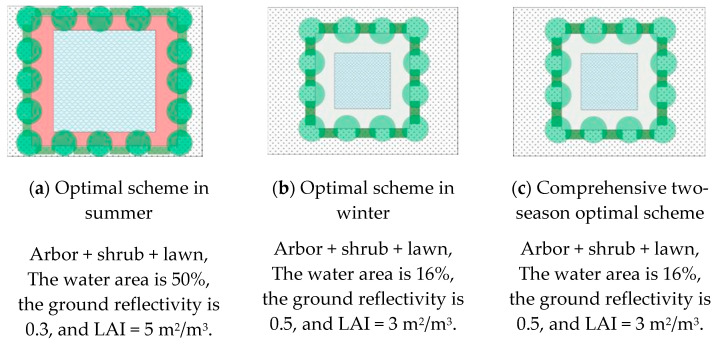
Plan diagram of optimization results of test scheme the tex.

**Table 1 ijerph-19-06303-t001:** Architectural layout characteristics of residential areas in Lhasa.

Building Layout Type			
Determinant Building Layout	Hybrid Building Layout	Closed Building Layout	Scattered Building Layout
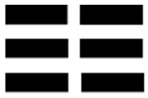	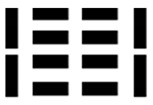	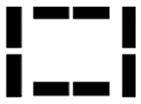	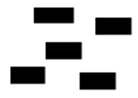
37	14	5	4

**Table 2 ijerph-19-06303-t002:** Landscape elements allocations characteristics of residential areas in Lhasa.

Landscape elements allocations	Layout form of greening		
	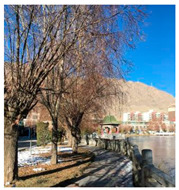 Arbor + lawn	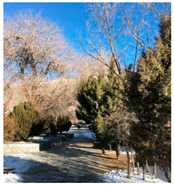 Arbor + shrub + lawn	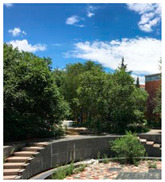 Shrub + lawn
	Underlying surface type		
	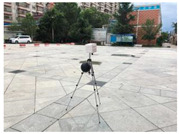 Slate floor	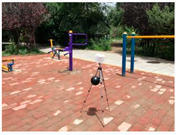 Permeable brick floor	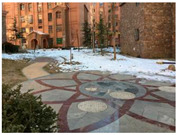 Marble floor
	Layout form of water body		
	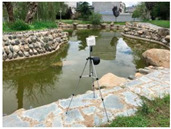 Still water	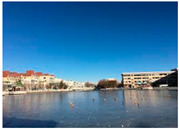 Still water	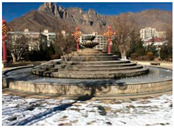 Fountain

**Table 3 ijerph-19-06303-t003:** Factors and working conditions of level setting.

Detailed Description of the Working Condition of the Underground Activity Ground			
**Working Condition**	**JX—A1**	**JX—A2**	**JX—A3**
Configuration description	1. 1/3 of the greening area is an arbor that is arranged around the activity site, and the rest is a lawn. 2. The height of the tree is 10 m, and the LAI = 3 m^2^/m^3^; The height of the grassland is 20 cm, and the LAD = 0.3 m^2^/m^3^.	1. 1/3 of the green area is shrubs that are arranged around the activity site, and the rest is a lawn. 2. The height of the shrubs is 2 m, and the LAD = 2.0 m^2^/m^3^; The height of the grassland is 20 cm, and LAD = 0.3 m^2^/m^3^.	1. In 1/3 of the area of greening, arbor:shrubs = 2:1, the arbor and the shrubs are arranged around the activity site at intervals, and the rest is a lawn. 2. The height of the trees is 10 m, LAI = 3 m^2^/m^3^; The height of the shrubs is 2 m, and the LAD = 2.0 m^2^/m^3^; The height of the grassland is 20 cm, and LAD = 0.3 m^2^/m^3^.
Detailed description of water area simulation conditions (the underground surface of the activity site is concrete)			
**Working condition**	**JX—B1**	**JX—B2**	**JX—B3**
Configuration description	Water area accounts for 16% of the green area	Water area accounts for 33% of the green area	Water area accounts for 50% of the green area
Reflectivity of different underlying surfaces (Reprinted with permission from Ref. [36]. 2002, Han, H.)			
**Working condition**	**JX—C1**	**JX—C2**	**JX—C3**
Floor materials	Concrete(reflectivity 0.2)	Red lime sand brick floor (reflectance 0.3)	White sintered granite (reflectance 0.5)
Detailed description of simulated working conditions of the LAI (the underground cushion surface of the movable site is concrete)			
**Working condition**	**JX—D1**	**JX—D2**	**JX—D3**
Floor materials	LAI = 1 m^2^/m^3^	LAI = 2 m^2^/m^3^	LAI = 3 3 m^2^/m^3^

**Table 4 ijerph-19-06303-t004:** Schematic diagram of factor level and plane of orthogonal test.

Factor Level	Green Configuration (A)	Water Area (B)	Ground Reflectance (C)	Leaf Area Index (D)
1	Arbor + lawn	16%	0.2	LAI = 1
2	Shrub + lawn	33%	0.3	LAI = 3
3	Arbor + shrub + lawn	50%	0.5	LAI = 5
Plane sketch	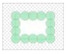	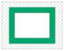	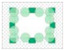	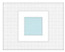	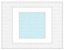	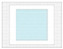	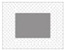	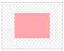	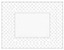	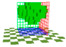	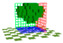	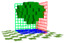
	A1	A2	A3	B1	B2	B3	C1	C2	C3	D1	D2	D3

**Table 5 ijerph-19-06303-t005:** Plane diagram of 27 test schemes.

27 Test Schemes
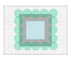	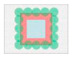	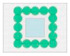	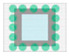	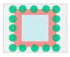	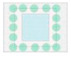	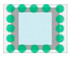	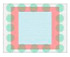	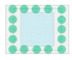
A1B1C1D1	A1B1C2D2	A1B1C3D3	A1B2C1D2	A1B2C2D3	A1B2C3D1	A1B3C1D3	A1B3C2D1	A1B3C3D2
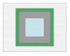	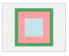	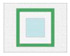	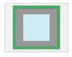		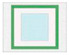	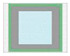	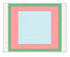	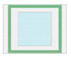
A2B1C1	A2B1C2	A2B1C3	A2B2C1	A2B2C2	A2B2C3	A2B3C1	A2B3C2	A2B3C3
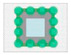	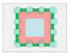	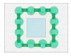	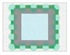		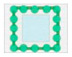	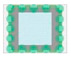	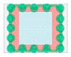	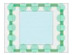
A3B1C1D3	A3B1C2D1	A3B1C3D2	A3B2C1D1	A3B2C2D2	A3B2C3D3	A3B3C1D2	A3B3C2D3	A3B3C3D1

**Table 6 ijerph-19-06303-t006:** Simulation time and boundary conditions.

Setting of Initial Conditions for Typical Meteorological Days in Summer			
Simulation date	06/21	Initial air temperature	18.9 °C
Start time	00:00	Relative humidity at 2 m	50%
Time of duration	24 h	Solar radiation adjustment coefficient	0.7
Wind speed/direction at 10 m	2.5 m/s, 290°	Cloudiness	4/8
Setting of Initial Conditions for Typical Meteorological Days in Winter			
Simulation date	01/21	Initial air temperature	1.7 °C
Start time	00:00	Relative humidity at 2 m	16%
Time of duration	24 h	Solar radiation adjustment coefficient	1.2
Wind speed/direction at 10 m	1.8 m/s, 90°	Cloudiness	4/8

**Table 7 ijerph-19-06303-t007:** Outdoor thermal comfort classification table.

PET (°C)	Thermal Comfort	Physiological Stress
<4	Very cold	Extreme cold stress
4–8	Cold	Severe cold stress
8–13	Cool	Moderate cold stress
13–18	Slightly cool	Mild cold stress
18–23	Comfort	No thermal stress
23–29	Slightly warmer	Mild heat stress
29–35	Warm	Moderate heat stress
35–41	Hot	Strong heat stress
>41	Very hot	Extreme heat stress

**Table 8 ijerph-19-06303-t008:** Results of orthogonal test in summer and winter.

Summer														
List		A	B	A × B	A × B	C	A × C	A × C	B × C	D	Empty	B × C	Empty	Empty
Morning	K1	137.21	135.93	132.30	134.01	130.22	132.61	133.77	132.50	135.01	133.31	132.00	133.51	132.61
	K2	134.96	134.28	134.51	132.31	132.87	132.81	132.77	133.70	130.80	132.71	133.15	133.01	133.30
	K3	126.85	128.80	132.21	132.70	135.93	133.61	132.48	132.81	133.21	133.01	133.86	132.50	133.11
	k1	15.25	15.10	14.70	14.89	14.20	14.73	14.86	14.72	15.00	14.81	14.67	14.83	14.73
	k2	15.00	14.92	14.95	14.70	14.76	14.76	14.75	14.86	14.53	14.75	14.79	14.78	14.81
	k3	14.09	14.31	14.69	14.74	15.10	14.85	14.72	14.76	14.80	14.78	14.87	14.72	14.79
	R	1.16	0.79	0.26	0.19	0.59	0.12	0.14	0.14	0.47	0.06	0.20	0.11	0.08
	ρj	48.47%	22.73%	2.77%	1.30%	13.36%	0.46%	0.74%	0.66%	7.26%	0.15%	1.48%	0.41%	0.21%
Afternoon	K1	292.32	297.54	290.79	290.70	294.21	289.89	290.34	289.62	293.40	289.98	291.15	290.25	290.16
	K2	301.05	289.26	288.36	289.98	287.28	289.80	289.89	291.24	290.97	289.71	289.80	289.71	289.71
	K3	276.30	282.87	290.52	288.99	288.09	290.07	289.44	288.81	285.30	289.98	288.72	289.71	289.80
	k1	32.48	33.06	32.31	32.30	32.69	32.21	32.26	32.18	32.60	32.22	32.35	32.25	32.24
	k2	33.45	32.14	32.04	32.22	31.92	32.20	32.21	32.36	32.33	32.19	32.20	32.19	32.19
	k3	30.70	31.43	32.28	32.11	32.01	32.23	32.16	32.09	31.70	32.22	32.08	32.19	32.20
	R	2.74	1.64	0.27	0.18	0.77	0.03	0.09	0.27	0.89	0.03	0.27	0.06	0.05
	ρj	63.00%	21.90%	0.72%	0.29%	5.81%	0.01%	0.08%	0.60%	6.91%	0.01%	0.61%	0.04%	0.03%
Night	K1	251.73	253.80	249.93	250.38	253.62	250.65	251.01	251.55	254.43	250.65	250.92	250.92	250.74
	K2	255.69	252.54	251.10	251.10	247.77	250.56	250.56	250.38	249.57	250.65	250.29	250.56	250.65
	K3	244.80	245.88	251.19	250.74	250.83	251.01	250.65	250.29	248.22	250.92	251.01	250.74	250.83
	k1	27.97	28.20	27.77	27.82	28.18	27.85	27.89	27.95	28.27	27.85	27.88	27.88	27.86
	k2	28.41	28.06	27.90	27.90	27.53	27.84	27.84	27.82	27.73	27.85	27.81	27.84	27.85
	k3	27.20	27.32	27.91	27.86	27.87	27.89	27.85	27.81	27.58	27.88	27.89	27.86	27.87
	R	1.21	0.87	0.14	0.08	0.65	0.05	0.05	0.14	0.69	0.03	0.08	0.04	0.02
	ρj	43.94%	25.97%	0.72%	0.20%	12.43%	0.13%	0.07%	0.78%	15.43%	0.03%	0.26%	0.03%	0.01%
**Winter**														
**List**		**A**	**B**	**A × B**	**A × B**	**C**	**A × C**	**A × C**	**B × C**	**D**	**Empty**	**B × C**	**Empty**	**Empty**
Morning	K1	−13.95	−8.55	−10.62	−10.62	−10.53	−10.08	−10.17	−10.44	−11.52	−10.26	−10.26	−10.26	−10.26
	K2	−6.48	−10.71	−13.05	−10.35	−11.70	−10.35	−10.35	−10.98	−9.18	−10.35	−9.99	−10.17	−10.17
	K3	−10.35	−11.52	−9.81	−9.81	−8.46	−10.35	−10.26	−9.36	−10.08	−10.17	−10.53	−10.35	−10.26
	k1	−1.55	−0.95	−1.18	−1.18	−1.17	−1.12	−1.13	−1.16	−1.28	−1.14	−1.14	−1.14	−1.14
	k2	−0.72	−1.19	−1.45	−1.15	−1.30	−1.15	−1.15	−1.22	−1.02	−1.15	−1.11	−1.13	−1.13
	k3	−1.15	−1.28	−1.09	−1.09	−0.94	−1.15	−1.14	−1.04	−1.12	−1.13	−1.17	−1.15	−1.14
	R	0.83	0.33	0.09	0.09	0.36	0.03	0.02	0.18	0.26	0.02	0.07	0.02	0.01
	ρj	64.82%	10.70%	0.84%	0.84%	12.38%	0.08%	0.06%	3.15%	6.63%	0.04%	0.42%	0.04%	0.00%
Afternoon	K1	123.12	130.50	126.81	127.44	125.73	126.99	127.35	126.45	127.80	127.17	126.99	126.81	127.35
	K2	124.11	126.54	127.08	127.17	127.35	127.08	127.08	127.98	127.89	126.90	126.27	127.26	126.81
	K3	134.28	124.29	127.53	126.72	128.25	127.26	126.90	126.90	125.73	127.35	127.98	127.44	127.17
	k1	13.68	14.50	14.09	14.16	13.97	14.11	14.15	14.05	14.20	14.13	14.11	14.09	14.15
	k2	13.79	14.06	14.12	14.13	14.15	14.12	14.12	14.22	14.21	14.10	14.03	14.14	14.09
	k3	14.92	13.81	14.17	14.08	14.25	14.14	14.10	14.10	13.97	14.15	14.22	14.16	14.13
	R	1.24	0.69	0.08	0.08	0.28	0.03	0.05	0.17	0.24	0.05	0.19	0.07	0.05
	ρj	71.98%	18.63%	0.25%	0.25%	3.03%	0.03%	0.08%	1.18%	2.78%	0.08%	1.43%	0.17%	0.08%

**Table 9 ijerph-19-06303-t009:** Analysis of variance results of various factors in summer and winter.

Time	Source	Calibration Model	Intercept	A	B	C	D	A × B	A × C	B × C	
Summer morning	F	38.722	338,776.47	189.144	88.739	52.136	28.34	7.915	2.339	4.175	R^2^ = 0.992 (adjust: R^2^ = 0.967)
	Sig.	0	0	0	0	0	0.001	0.014	0.169	0.059	
Summer noon	F	384.031	3,886,834.995	2421.293	841.525	223.457	265.478	19.474	1.589	23.218	R^2^ = 0.999 (adjust: R^2^ = 0. 997)
	Sig.	0	0	0	0	0	0	0.001	0.291	0.001	
Summer evening	F	450.938	12,446,515.42	2019.457	1241.738	507.76	678.754	11.33	2.603	16.904	R^2^ = 0.999 (adjust: R^2^= 0.997)
	Sig.	0	0	0	0	0	0	0.006	0.142	0.002	
Winter morning	F	336.627	49,570.309	2181.209	363.618	419.126	223.733	27.673	2.51	59.11	R^2^ = 0.999 (adjust: R^2^ = 0.996)
	Sig.	0	0	0	0	0	0	0.001	0.151	0	
Winter noon	F	74.738	681,196.485	540.229	139.769	22.512	20.994	1.79	0.481	9.665	R^2^ = 0.996 (adjust: R^2^ = 0.983)
	Sig.	0	0	0	0	0.002	0.002	0.249	0.75	0.009	

**Table 10 ijerph-19-06303-t010:** Calculation of interactions between A and B as well as B and C in summer and winter.

Summer								
		A1	A2	A3		B1	B2	B3
Morning	B1	**15.60**	15.45	14.25	C1	14.63	14.78	14.00
	B2	15.48	15.16	14.13	C2	15.18	14.77	14.33
	B3	14.66	14.37	13.91	C3	**15.50**	15.21	14.60
Afternoon	B1	33.53	34.03	31.63	C1	33.66	32.64	31.78
	B2	32.25	33.54	30.64	C2	32.92	31.88	**30.98**
	B3	31.67	32.77	**29.84**	C3	32.61	31.91	31.52
Night	B1	28.27	28.79	27.63	C1	28.63	28.37	27.54
	B2	28.25	28.63	27.31	C2	27.87	27.70	**27.12**
	B3	27.47	27.82	**26.67**	C3	28.20	28.12	27.31
**Winter**								
		**A1**	**A2**	**A3**		**B1**	**B2**	**B3**
Morning	B1	−1.44	**−0.49**	−0.93	C1	−1.00	−1.33	−1.17
	B2	−1.61	−0.76	−1.19	C2	−1.16	−1.25	−1.49
	B3	−1.58	−0.91	−1.34	C3	**−0.69**	−0.98	−1.24
Afternoon	B1	14.06	14.10	**15.35**	C1	14.27	14.11	13.55
	B2	13.61	13.79	14.77	C2	14.53	14.05	13.86
	B3	13.36	13.44	14.63	C3	**14.71**	14.03	14.05

**Table 11 ijerph-19-06303-t011:** Optimization calculation results for summer and winter.

Test Number	PET_s_	PET_w_	β	Test Number	PET_s_	PET_w_	β
1	18.71	6.91	2.71	15	18.27	7.46	2.45
2	17.98	7.07	2.54	16	18.23	6.85	2.66
3	17.84	7.26	2.46	17	17.71	7.02	2.52
4	18.12	6.84	2.65	18	17.63	7.08	2.49
5	17.51	6.64	2.63	19	17.63	7.77	2.27
6	18.11	6.80	2.66	20	17.60	7.90	2.23
7	17.45	6.50	2.68	21	17.43	8.40	2.08
8	17.53	6.54	2.68	22	17.70	7.57	2.34
9	17.50	6.86	2.55	23	16.98	7.63	2.22
10	18.73	7.50	2.50	24	17.01	7.56	2.25
11	18.11	7.43	2.44	25	16.97	7.42	2.29
12	18.61	7.68	2.42	26	16.40	7.29	2.25
13	18.31	7.07	2.59	27	17.07	7.62	2.24
14	18.39	7.20	2.55				

## Data Availability

The data presented in this study are available on request from the corresponding author.

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
