# Peer review of "Optimization Design of the Landscape Elements in the Lhasa Residential Area Driven by an Orthogonal Experiment and a Numerical Simulation"

_ijerph, 2022, doi:10.3390/ijerph19106303_

Round 1

Reviewer 1 Report

The paper presents a model for different scenarios for a city in the Tibetan Plateau, considering the physiological equivalent temperature (PET). The authors present the high demand for the comfort of the outdoor spaces in the city as an issue to be investigated through their proposed methodology. The highlighted results are that a combination of factors would be beneficial for environmental comfort in the city in both summer and winter.

The study is primarily exploratory, considering some baselines from the Chinese context of planning and designing housing settlements. The literature review is short in terms of the diversity of perspectives and concepts. The contribution is mainly focused on presenting the results from the model, lacking in detailing the justifications and broadening the discussions. The authors should consider including citations from the same subject that were published in different contexts/countries. It would support the improvement of the discussion section. The short number of references and years of publication is also something that should be improved by the authors. The paper could contribute to discussions on green infrastructure and resilient cities subjects.

If it does not expand the number of pages too much, another suggestion for the authors is to consider inserting a map of the city, highlighting the urban layout and actual vegetation cover and water bodies in the city. It would support the readers to get an idea of how the city is structured.

In the 2.2 section, the results of a questionnaire survey are presented. The details of the survey are not presented and it is not quite clear to the reader what questions were asked to the interviewees.

Citations and references to figures in the body of the text are not in the right formatting. It must be checked throughout the manuscript.

On line 117 it is mentioned previous research which is not cited or referenced in the text.

Author Response

Response to Reviewer 1 Comments

Point 1:

The study is primarily exploratory, considering some baselines from the Chinese context of planning and designing housing settlements. The literature review is short in terms of the diversity of perspectives and concepts. The contribution is mainly focused on presenting the results from the model, lacking in detailing the justifications and broadening the discussions. The authors should consider including citations from the same subject that were published in different contexts/countries. It would support the improvement of the discussion section. The short number of references and years of publication is also something that should be improved by the authors. The paper could contribute to discussions on green infrastructure and resilient cities subjects.

Response 1: Thank you very much for your valuable comments, which are very helpful to the improvement of our paper. In response to your comments, we have made the following three modifications:

  • The introduction is divided into 1.1 background and 1.2 literature review, and the summary of 1.2 literature review emphasizes the differences of landscape elements in different climate areas on outdoor thermal comfort, but there are few relevant studies on high-altitude cold climate areas at present. The importance and necessity of this study are pointed out;
  • In the literature review part, we interpret and analyze the relevant literature, especially the literature in recent years, and list it in the references. At present, there are 43 references, and nearly 1/2 of them are the literature in recent five years.
  • we also added a part " 4. Discussion". It compares the impact of landscape elements and their interaction on thermal comfort between other studies and this study, points out the research results of this study on the optimal design of landscape element allocation in Lhasa City, a high-altitude cold climate area, and emphasizes the original contribution of this study. In addition, it also points out the topics that we should focus on and deepen in the future research.

Point 2:

If it does not expand the number of pages too much, another suggestion for the authors is to consider inserting a map of the city, highlighting the urban layout and actual vegetation cover and water bodies in the city. It would support the readers to get an idea of how the city is structured.

Response 2: Thank you for your valuable advice. We have added Figure 1 in part 2.1, highlighting the urban layout, large-area vegetation cover and water bodies in the city.

Point 3:

In the 2.2 section, the results of a questionnaire survey are presented. The details of the survey are not presented and it is not quite clear to the reader what questions were asked to the interviewees.

Response 3: Thank you very much for your comments, for which we have added in 2.2 Characteristics of residents' behavior habits about the actual distribution of the questionnaire and what it contains. The details can be seen on line 150 to 158.

Point 4:

Citations and references to figures in the body of the text are not in the right formatting. It must be checked throughout the manuscript.

Response 4: Thank you very much for the reminder, we have read through the whole article and checked the citation and reference formats in the body text one by one.

Point 5:

On line 117 it is mentioned previous research which is not cited or referenced in the text.

Response 5: Thank you for asking the question here, here is our description error, which has been modified for this issue.

Response 6:For grammar and spelling errors, we have polished the manuscript through Letpub.

Reviewer 2 Report

The authors submitted an interesting manuscript dealing with landscape optimization design of outdoor space in the residential area of Lhasa. With increasing changes in the climate system, this research could provide some insights on adaptation of these changes. However, the manuscript should be revised before it could be considered for publication. Some comments and suggestions are listed below:

Lines 28-50: In the section of Introduction, the authors should clearly provide the aim of the study. They should provide a short introduction on the methods adopted to conduct this study.

Line 64: Please provide a sharper image for the Figure 1.

Line 78: For the Figure, please provide more description for each outdoor space provide on this Figure.

Line 87: What do you mean by the main content ?

Line 89: Please provide the version and the manufacturer of ENVI-met software.  

Lines 86-94: The research framework provide is not sufficient. Please provide more details on the elements presented on the Figure 4.

Lines 104-106: on the Figure 4, we can't find A and B. Please check if you should mention Figure 5.

Line 115: Please provide the reference for leaf area indexes or provide its formula. Please indicate how it was obtained.

Line 117: The Table 1 should be near it is cited in the text. Please relocate the Table 1 from Lines 134-135 and put it between lines 119 and 120.

Line 124: The Table 1 should be near it is cited in the text. Please relocate the Table 1 from Lines 134-135 and put it between lines 119 and 120. Please also check for the Tables 2, 3

Line 189: Please provide the version and the manufacturer of SPSS software.  

Line 281: for the Figure 7, please provide the legend for different colors presented on this Figure.

Author Response

Response to Reviewer 2 Comments

Point 1:

Lines 28-50: In the section of Introduction, the authors should clearly provide the aim of the study. They should provide a short introduction on the methods adopted to conduct this study.

Response 1: Thank you for your valuable advice, we are in 1.2. The Literature review concludes with the addition of the methodology and purpose of the study. The details can be seen on line 102 to 118.

Point 2:

Line 64: Please provide a sharper image for the Figure 1.

Response 2: Thank you very much for the reminder, Figure 1 has been replaced by a much clearer picture.

Point 3:

Line 78: For the Figure, please provide more description for each outdoor space provide on this Figure.

Response 3: Thanks for your valuable comments, for the text description of the picture, we have replaced two more appropriate pictures (Figure 3), Table 1 and Table 2 have been added to 2.1 Climate Conditions and Residence in Lhasa, to describe the architectural layout and landscape characteristics of Lhasa residential area.

Point 4:

Line 87: What do you mean by the main content ?

Response 4: Thank you for your question here, here is our language error, we have changed "The main content of this paper" to "The research framework of this study".

Point 5:

Line 89: Please provide the version and the manufacturer of ENVI-met software.

Response 5: Thank you for your valuable feedback, we have provided the version and manufacturer of The Envimet software in this article. " ENVI-met, a three-dimensional urban microclimate simulation software jointly developed by Michael Bruse and Heriberfleet of Bochum University in Germany, was used for numerical simulation. This study used ENVI-met version v4.4.3 for the calculation to study the influence degree and the order of different landscape elements and their interaction with the thermal environment in residential areas. ".

Point 6:

Lines 86-94: The research framework provide is not sufficient. Please provide more details on the elements presented on the Figure 4.

Response 6: Thanks for your valuable feedback, we have optimized Figure 4.

Point 7:

Lines 104-106: on the Figure 4, we can't find A and B. Please check if you should mention Figure 5.

Response 7: Thank you for your valuable comment, here is our description error, which has been corrected for this issue.

Point 8:

Line 115: Please provide the reference for leaf area indexes or provide its formula. Please indicate how it was obtained.

Response 8: Thank you for your valuable feedback, we have added references to the leaf area index and described the basis for determining the leaf area index. In this research, we studied the impact of tree leaf area index (LAI) on outdoor thermal environment. According to the literature (Sang.G.) , the LAI of most plants is between 0 −6m2 / m3, Therefore, we set the LAI to 1, 3 and 5.

Point 9:

Line 117: The Table 1 should be near it is cited in the text. Please relocate the Table 1 from Lines 134-135 and put it between lines 119 and 120.

Response 9: Thanks for your valuable comments, we have placed Table 1 near it is cited in the text.

Point 10:

Line 124: The Table 1 should be near it is cited in the text. Please relocate the Table 1 from Lines 134-135 and put it between lines 119 and 120. Please also check for the Tables 2, 3

Response 10: Thanks for your valuable comments, we have placed Table 2 and Table3 near it is cited in the text.

Point 11:

Line 189: Please provide the version and the manufacturer of SPSS software. 

Response 11: Thank you for your valuable feedback, we have provided the version and manufacturer of SPSS software in the article. “The statistical product service solutions (SPSS) statistical analysis software from IBM was used to analyze the significance of each influencing factor. The version of IBM SPSS statistics. 22 was adopted in this research. ”

Point 12:

Line 281: for the Figure 7, please provide the legend for different colors presented on this Figure.

Response 12: Thank you for pointing out the shortcomings here, we have supplemented the parameter settings for landscape feature configuration under each pilot scenario in Figure 7.

Response 13:For grammar and spelling errors, we have polished the manuscript through Letpub.

Round 2

Reviewer 2 Report

No further comments